# Generalized Structure Functions and Multifractal Detrended Fluctuation Analysis Applied to Vegetation Index Time Series: An Arid Rangeland Study

**DOI:** 10.3390/e23050576

**Published:** 2021-05-07

**Authors:** Ernesto Sanz, Antonio Saa-Requejo, Carlos H. Díaz-Ambrona, Margarita Ruiz-Ramos, Alfredo Rodríguez, Eva Iglesias, Paloma Esteve, Bárbara Soriano, Ana M. Tarquis

**Affiliations:** 1CEIGRAM, Universidad Politécnica de Madrid, 28040 Madrid, Spain; antonio.saa@upm.es (A.S.-R.); carlosgregorio.hernandez@upm.es (C.H.D.-A.); margarita.ruiz.ramos@upm.es (M.R.-R.); alfredo.rodriguez@uclm.es (A.R.); eva.iglesias@upm.es (E.I.); paloma.esteve@upm.es (P.E.); barbara.soriano@upm.es (B.S.); anamaria.tarquis@upm.es (A.M.T.); 2Grupo de Sistemas Complejos, Universidad Politécnica de Madrid, 28040 Madrid, Spain; 3Evaluación de Recursos Naturales, ETSI Agronómica, Alimentaria y Biosistemas, Universidad Politécnica de Madrid, 28040 Madrid, Spain; 4AgSystems, ETSI Agronómica, Alimentaria y Biosistemas, Universidad Politécnica de Madrid, 28040 Madrid, Spain; 5Departamento de Análisis Económico y Finanzas, Universidad de Castilla-La Mancha, 45071 Toledo, Spain; 6Economía Agraria y Gestión de los Recursos Naturales, ETSI Agronómica Alimentaria y Biosistemas, Universidad Politécnica de Madrid, 28040 Madrid, Spain

**Keywords:** NDVI, multiscaling, generalized structure function, detrended fluctuation analysis, grassland

## Abstract

Estimates suggest that more than 70% of the world’s rangelands are degraded. The Normalized Difference Vegetation Index (NDVI) is commonly used by ecologists and agriculturalists to monitor vegetation and contribute to more sustainable rangeland management. This paper aims to explore the scaling character of NDVI and NDVI anomaly (NDVIa) time series by applying three fractal analyses: generalized structure function (GSF), multifractal detrended fluctuation analysis (MF-DFA), and Hurst index (HI). The study was conducted in four study areas in Southeastern Spain. Results suggest a multifractal character influenced by different land uses and spatial diversity. MF-DFA indicated an antipersistent character in study areas, while GSF and HI results indicated a persistent character. Different behaviors of generalized Hurst and scaling exponents were found between herbaceous and tree dominated areas. MF-DFA and surrogate and shuffle series allow us to study multifractal sources, reflecting the importance of long-range correlations in these areas. Two types of long-range correlation appear to be in place due to short-term memory reflecting seasonality and longer-term memory based on a time scale of a year or longer. The comparison of these series also provides us with a differentiating profile to distinguish among our four study areas that can improve land use and risk management in arid rangelands.

## 1. Introduction

Many rangelands are suffering severe degradation processes. The tailored monitoring of vegetation to inform the sustainable management of these areas will prevent their degradation [1,2,3]. New tools and metrics use complexity to understand and predict natural systems’ behavior and improve monitoring and management programs. In the past decades, advances suggest that complex-systems science can develop prediction frameworks with metrics that explain the underlying causes of spatiotemporal dynamics [4].

Remote sensing and satellite monitoring methods are commonly used to study ecology and agriculture [5,6,7,8]. Among different satellite indices, the Normalized Difference Vegetation Index (NDVI) is the most widely used vegetation index by rangeland ecologists [9]. The NDVI reveals a good correlation with biomass of different vegetation types in arid and semi-arid areas [10,11,12]. NDVI anomaly (NDVIa, the relative difference of NDVI to long-term averages of the index) is another index proven useful for identifying drought and water stress on vegetation; therefore, it is suited to study arid and semi-arid ecosystems [13,14].

Ecosystems are complex dynamical systems composed of different communities driven by various processes operating at different spatial and temporal scales [15,16]. Ecosystems also show a self-organizing capacity, as they are non-linear systems [17,18,19]. Therefore, non-linear techniques can be used to understand plant community spatio-temporal dynamics [20,21,22]. The mathematical properties of both temporally and spatially complex systems are often fractal [23]. Alados et al. and Saravia et al. [24,25,26] demonstrated a negative correlation between plant species diversity and fractal dimension. Most papers studying vegetation series have focused on monofractal techniques calculating the Hurst index (HI), a persistence test, usually estimated by the rescaled range (R/S) method [27,28,29]. Li et al. [30] began to calculate the HI using the detrended fluctuation analysis (DFA), as it is a more robust method to detect the scaling behavior in time series. Many of these works focus on studying how different HI values are related to different vegetation dynamics, and can lead to a further understanding of the interaction between their different components [31].

Besides HI, several multifractal analyses have been used to provide a more in-depth comparison of time series. For example, the generalized structure function (GSF) [32] and multifractal detrended fluctuation analysis (MF-DFA) [33] focus on measuring variations of the moments of the absolute difference of their values at different scales. These methods provide several metrics dependent on the order of *q* moments, generalized Hurst exponents (*H*(*q*)), and others such as scaling exponent (*ζ*(*q*)) that differ in the calculation for each method, as explained below. When *q* = 2, the generalized Hurst exponent (*H*(*2*)) represents persistence, similarly to HI, but from a multifractal technique. The GSF has been used to study vegetation [34] and other geophysical data [35]. More recently, MF-DFA has been used to study the long-term ecosystem dynamics at a large scale [36,37,38,39] and compare the dynamics of areas affected and unaffected by fire [40]. MF-DFA allows the study of multiscaling on vegetation and the detection of whether it is related to long-term correlations or broad probability density functions. Studying the difference of multiscaling between different areas can further support our understanding of vegetation dynamics and its interaction with other components [41]. Interpretations of the monofractal and multifractal analysis of landscapes have been used to inform policymakers. Different studies have been conducted to predict vegetation dynamics [42,43], while others have developed tools to evaluate current management practices [36,44,45].

In this paper, we aim to characterize the multifractality of selected representative rangelands from a semi-arid environment using parameters that can be derived from the GSF and MF-DFA. For this, NDVI and NDVIa series analysis are used to study the vegetation dynamics of several rangelands that present different land uses. Understanding the multifractal character of rangelands will enable the characterization of their dynamics, improving their management accordingly, and provide new insight to assess risk that can be used to design rangeland index insurances.

## 2. Area of Study, Data Collection and Methods

### 2.1. Area of Study

The study areas were rangelands located in the Murcia region, in the southeast of Spain. The area has a Mediterranean arid climate with annual precipitation below 300 mm, although there are regional variations [46]. Four square areas of 2.5 km per side were selected for this study. Each of them is situated close to a meteorological station, located in three different agricultural counties of Murcia: Northeast, Segura River, and Northwest (Figure 1).

The description of the study areas is as follows: Area 1 (A1) is mainly covered by rainfed cereal crops and stubble, Area 2 (A2) is almost entirely covered by mixed rainfed croplands used for stubble grazing with some grassland and scrubland, Area 3 (A3) has a larger grassy area mixed with shrubs and few forested areas surrounded by irrigated mixed and tree crops, and Area 4 (A4) is mainly covered by coniferous woodland with mixed rainfed crops on small patches. Satellite images and the NDVI averages for all years are shown in Figure 2.

### 2.2. NDVI Data Collection

MOD09A1.006 products were collected from AppEEARS [47], downloading the *RED* (band 1) and *NIR* (band 2) values as well as the quality bands for the study areas. This product has a 250 × 250 m^2^ spatial resolution, implying 132 pixels for each area, and an 8 day temporal resolution from the beginning of 2002 to 2019, which makes 18 years of data series. For each pixel series, R software was used to calculate the *NDVI*, using the following formula:(1)$$NDVI=100\times \frac{NIR-RED}{NIR+RED}$$

The possible *NDVI* values range from 0 to 100. The *NDVI* values were then checked for quality. If the data were not categorized as ideal quality according to the quality band from AppEEARS [47], they were deleted; less than 0.01% were deleted for every area. The gaps were filled using running averages with a gap interval of seven dates. Then, the time series were smoothed to reduce noise due to cloud contamination and associated shadows. The Savitzky–Golay method [48] was used, with a window size of 9, selected based on the best-fitted outputs. The *NDVI* anomaly (*NDVIa*) was computed following Anyamba and Tucker [49]:(2)$$NDVIa=NDVI-\mu_{NDVI}$$
where ${µ}_{NDVI}$ is the average of all samples in all available years measured in the same calendar date. In this new time series, the seasonal variations of NDVI were eliminated.

### 2.3. Methods

We applied a combination of methods to study NDVI series that allows an in-depth understanding of the NDVI performance: the NDVI temporal trend, where we studied the presence of changes and trends in the dynamics of our time series; the HI, a monofractal technique to analyze the persistence of our time series; as well as the GSF and MF-DFA, multifractal techniques that use different mathematical approaches to study a multitude of parameters (persistence, among others) of time series to see how they scale differently. The major distinction between the last two techniques is that MF-DFA, unlike the GSF, has a detrending process that eliminates the temporal trend detected in our time series that influenced the multifractality analysis. Furthermore, we used MF-DFA to study the different sources of multifractality.

#### 2.3.1. Temporal Trend Analysis

The Mann–Kendall test [45,46] and Pettitt change test [47] were used to explore the NDVI series further. The Mann–Kendall test shows whether the trends are increasing or decreasing. Moreover, the Pettitt change test is used to analyze any abrupt changes in the trend of the series [44]. The package “trend” (version 1.1.2, in R Software) was used to apply both tests.

#### 2.3.2. Hurst Index

To analyze the persistence of the NDVI in each area, the HI [50] was calculated using the package “pracma” (version 1.9.9, [51]) in R software. We used the corrected Hurst index. This index splits the time series into subseries, with τ indicating the subseries number. It normalizes the subseries, subtracting each subseries its average. Secondly, it calculates the cumulative series range (*R*(*τ*)) for each subseries. This range is divided by the standard deviation (*S*(*τ*)) of each subseries. The Hurst index (HI) is then calculated as *H*, using the following formula and averaging each subseries with c as a constant.
(3)$$\frac{R\left(\tau \right)}{S\left(\tau \right)}={\left(c\tau \right)}^{H}$$

When the HI or the generalized Hurst exponent for q = 2 (*H*(2)) stay above 0.5, this indicates that the series is persistent. If HI and *H*(2) fall below 0.5, this means that the series is antipersistent. Brownian motion or random walk would show an HI of 0.5.

#### 2.3.3. Generalized Structure Function

The GSF is used to characterize the scaling behavior of non-overlapping fluctuation at different scale increments. For non-stationary processes, a GSF of order *q* is defined as the *q* th moment of initial values *x*(*i*) increments. The equation is:(4)$$M_{q}\left(\tau \right)\equiv \langle {\left|x\left(i+\tau \right)-x\left(i\right)\right|}^{q}\rangle$$
where *i* denotes the *i* th data point and ⟨ ⟩ represents the ensemble average, and τ is the lag time (i.e., *i* ± τ representing the ${\left(i\pm \tau \right)}^{\mathrm{th}}$ data point). GSFs are generalized variogram functions [52,53,54], being particularly evident from Equation (4) for *q* = 2. This would result in a variogram of second order, which is frequently used in geostatistics. In general, *q* may be any real number, either positive or negative. However, there are divergence problems inherent to a negative-order exponent, so computations are best restricted to positive real numbers. We use positive *q* values up to 4 in this work to reduce increasing errors related to higher-order statistical moments [55]. If the process *x(i)* is scale-invariant and self-similar or self-affine over some range of time lags $\tau_{min}\leq \tau \leq \tau_{max}$, then the *q* th-order structure function is expected to scale as:(5)$$Mq\left(\tau \right)\equiv C_{q}\tau^{\zeta \left(q\right)}\approx \tau^{\zeta \left(q\right)}$$
where *C_q_* can be a function of τ, which varies more slowly than any power of τ, and *q* is the exponent of the structure function. The scaling exponent, *ζ*(*q*), is a monotonically non-decreasing function of *q* if *x(i)* has absolute bounds [56,57]. *ζ(q)* is calculated for the time scales where the fluctuation functions increase linearly, with lags starting at eight days (time between NDVI collections).

The behavior described by Equations (4) and (5) is called “multiscaling” because each statistical moment scales with a different exponent. Therefore, a hierarchy of exponents can be defined using *q* as:(6)$$H\left(q\right)=\frac{\zeta \left(q\right)}{q}$$
where *H*(*q*) is the generalized Hurst exponent [55] and is used to calculate *ΔH* as *H*(*0.25*)−*H*(*4*). *ΔH* is used as a measure of multifractality to compare each area. For a Brownian motion, given that it has no q-dependency (*H(q)* = 0.5), it would have Δ*H* = 0.

#### 2.3.4. Multifractal Detrended Fluctuation Analysis

The main feature of multifractals is that a high variability characterizes them over wide ranges of temporal or spatial scales associated with intermittent fluctuations and long-range power-law correlations. To undertake a multifractal analysis, Kantelhardt et al. [33] developed multifractal detrended fluctuation analysis (MF-DFA).

MF-DFA operates on *x*(*i*), where i=1,2,…,N and *N* is the series’ length. We represent the mean value with $\bar{x}$. We assume that *x(i)* are increments of a random walk process around the average $\bar{x}$. Therefore, the integration of the signal gives the profile:(7)$$y\left(i\right)=\overset{i}{\underset{k=1}{\sum }}\left[x\left(k\right)-\bar{x}\right]$$

Furthermore, the integration will reduce the level of measurement noise present in observational and finite records. Next, the integrated series is divided into *N*_s_ = int (*N*/s) non-overlapping segments of equal length *s*. We then calculate the local trend for each of the *N_s_* segments by a least-squares fit. Finally, we determine the variance:(8)$$F^{2}\left(s,\nu \right)=\frac{1}{s}\overset{s}{\underset{i=1}{\sum }}{\left\{y_{\left[\left(\nu -1\right)s+i\right]}-y_{\nu }\left(i\right)\right\}}^{2}$$
for each segment ν, where $\nu =1,\ldots ,N_{S}$. Here, $y_{\nu }$(*i*) is the fitting curve in segment ν. In this case, a linear regression was used in the fitting procedure. After detrending the series, we averaged over all segments to obtain the *q* th-order fluctuation function:(9)$$F_{q}\left(s\right)={\left\{\frac{1}{2N_{s}}\sum {\left[F^{2}\left(s,\nu \right)\right]}^{\frac{q}{2}}\right\}}^{\frac{1}{q}}$$
where in general, the index variable *q* can take any real value except zero. In this case, we selected real positive numbers.

Repeating the procedure described above for several time scales *s*, *F_q_*(*s*) will increase with increasing *s*. We can determine the fluctuation functions’ scaling behavior by analyzing the log–log plots of *F_q_*(*s*) versus *s* for each value of *q*. If the series *x*(*i*) is long-range power-law correlated, *F_q_*(*s*) increases for large values of *s* as a power law:(10)$$F_{q}\left(s\right)\;\propto s^{H\left(q\right)}$$
(11)$$H\left(q\right)\propto \;\frac{\log\left(F_{q}\left(s\right)\right)}{\log\left(s\right)}$$

*H(q)* is the generalized Hurst exponent as a function of *q. H(q)* is calculated for MF-DFA using the slope of log(*F_q_(s)*) vs. log(*s*) for the time scales, where the fluctuation functions increase linearly at a logarithmic scale (Equation (11)). Calculation started at 32 days and ran until 512 days. Observing Equations (9) and (10), when *q* = 2, the equation will be:(12)$$F_{2}\left(s\right)=\sqrt{\frac{1}{2N_{s}}\sum F^{2}\left(s,\nu \right)}$$
(13)$$F_{2}\left(s\right)\;\propto s^{H\left(2\right)}$$

Therefore, *H(2)* corresponds to the Hurst index estimated using MF-DFA as used by Li et al. (2017).

As mentioned above, a monofractal series with compact support is characterized by *H(q)* independent of q. Different scaling of small and large fluctuations will yield a significant dependence of *H(q)* on q, producing a higher *ΔH*, calculated as *H*(*0.25*)–*H*(*4*), as previously stated for the GSF. The difference in scaling increases with increasing dependency. Once *H(q)* is calculated, the scaling exponent (*ζ*(*q*)) is derived from the expression *H(q)/q* [33], as opposed to the GSF method, where *ζ*(*q*) is first calculated using Equation (5) and then *H(q)* is calculated.

There are two sources of multifractality: (i) due to a broad probability density function and (ii) due to different long-range correlations [33]. To test the study areas’ multifractality sources, we used the shuffle series to eliminate any temporal correlation. If the shuffle series had any multifractality, it would be due to a broad probability density function (pdf). The shuffle series were obtained using a random array of the length of our time series. We ordered our time series, matching the order of the random array. To test long-range correlations, we used surrogate series (or phase-randomized series). Surrogate series were calculated using iterated amplitude adjusted Fourier transform (IAAFT) [58,59]. If the surrogate series exhibited multifractality, it would be due to long-range correlations. Ten surrogate and shuffle series were calculated and averaged to compare them with the original series, based on previous studies [37,58]. The difference between the *ΔH* of the original and shuffle series (Hpdf) is quantitatively related to the influence of broad pdf, and the difference of *ΔH* of the original and surrogate series (Hcor) is related to the influence of long-range correlation [60].

## 3. Results

### 3.1. Temporal Trend Analysis

Several trends were detected in both the original NDVI and NDVIa for the study areas (Figure 3). When the Mann–Kendall test was used, all four areas exhibited a significant trend: A1 and A3 decreased, and A2 and A4 increased (Table 1). The Pettitt test found a shift in the trend in the four areas: A1 and A2 in the autumn of 2006; A3 in the autumn of 2007; and A4 in the autumn of 2008. All areas showed a shift at similar times between the NDVI and NDVIa. The only exception was A3, where the shift in the NDVIa occurred 2 years later than in the NDVI. This is likely due to the higher heterogeneity in this area (Figure 3). The results of the Mann–Kendal and Pettitt tests indicate a changing temporal trend in all series.

### 3.2. Hurst Index

The HI (Table 2), ranging between 0.91 (NDVIa) and 0.68 (NDVI), indicated a persistent character for the NDVI and NDVIa series in the study areas. However, some differences were found between the original NDVI and the NDVIa. The most significant changes between the HIs of the NDVI and NDVIa were found in A4, where persistence increased by roughly 0.2 when the NDVIa was used. Meanwhile, smaller increases, between 0.02 and 0.12, were observed in the other areas. Considering the NDVI, A2 was the most persistent (the highest HI), followed by A3, A4, and A1. Nevertheless, when the NDVIa was analyzed, a different order emerged, and A4 appeared as the most persistent. Its more extensive tree cover and woodland nature are less prone to a high variability, showing a higher persistence. On the other hand, the other areas had similar values and maintained the same order as the NDVI (A2, A3, and then A1).

### 3.3. Generalized Structure Function

The fluctuation functions for any moment order q reveals the multifractality of a series by its power-law behavior. Plotting the fluctuation function in log–log scale shows the multifractality by its linear behavior from 0.25 to 4. The scaling exponent was calculated at four different time scales (τ): 8, 16, 32, and 64 days (Figure 4, showing only A3 for illustrative purposes). Observing the scaling exponent for the NDVI, in Figure 5 A4 appears on the top of the plot as the least multifractal, while A1 was the most multifractal and A2 and A3 were in between. For the NDVIa, the scaling exponent showed slight differences between A2, A3, and A4, while A1 appeared further below the others.

The results of the generalized Hurst exponents with the GSF produced similar results when compared to R/S analysis. The HI and *H*(*2*) from this analysis showed minor differences when applied to the NDVI series, with disparities ranging from 0.007 to 0.12 (Table 2). These differences were higher when NDVIa series were used, with a difference ranging from 0.12 to 0.3. The NDVI series with the GSF was more persistent than the NDVIa series. In both cases, the analysis revealed a persistent character in all four areas. Comparing the NDVI series of the four areas (Figure 6), A1 appeared as the least persistent, A2 and A3 had a very similar pattern, and A4 showed a more persistent character than the other areas.

On the other hand, the NDVIa results were more similar among the areas. Area 2 appeared as the most persistent, A3 and A4 were in the middle, and A1 was the least persistent. Furthermore, A1 and A2 showed a similar pattern, while A3 and A4 presented a more constant profile that was very similar between them.

The *ΔH* for the NDVI with GSF had a minimum of 0.063 and a maximum of 0.116 (Table 3). There were differences between the NDVI and NDVIa. Whilst A4 and A1 were the most multifractal with the NDVI, followed by A3 and then A2, a different order emerged with the NDVIa. A1 and A2 showed the highest *ΔH* values with 0.137 and 0.116, respectively. On the other hand, A3 and A4 exhibited small values (0.015 and 0.012, respectively). These results match the different patterns between the NDVI and NDVIa observed in the generalized Hurst exponents for the GSF.

### 3.4. Multifractal Detrended Fluctuation Analysis

The fluctuation functions revealed multifractality of the series when analyzed with the GSF and MF-DFA. The generalized Hurst exponents for MF-DFA were calculated using four and five different time scales (*s*) for the NDVI and NDVIa, respectively (Figure 7, only showing A2 for illustrative purposes). Due to the trends in most series, we used MF-DFA to avoid the effects on generalized Hurst exponents and compare the different series for fractal character and persistence or antipersistence. In opposition to the GSF results, the generalized Hurst exponents of MF-DFA mainly indicated an antipersistent character for both indices. For the NDVI, from top to bottom, A4 appeared as the most persistent area, A2 and A3 appeared in the middle, and A1 showed an antipersistent character (Figure 8). All of them started above 0.5, but with the exception of A4, they all dropped below 0.5 for *q* ≥ 2, reflecting an antipersistent character. A more antipersistent plot emerged when the NDVIa was used—A1 and A2 started above 0.5 and immediately decreased to an antipersistent trend as *q* grew. A3 and A4 had similar patterns but with more antipersistent profiles. Moreover, the NDVI and NDVIa for A1 and A2, and for A3 and A4, had two different profiles, although for the NDVI each group kept the same pattern but at a different persistency level. The *ΔH* for MF-DFA, compared to the GSF, tended to have higher values, ranging from 0.206 to 0.409. The *ΔH* for MF-DFA decreased when the NDVIa was used for the study areas, with a more significant difference in A2 (Table 3). The common patterns mentioned above were reflected in their *ΔH*. A1 and A2 had a higher *ΔH*, and A3 and A4 had a lower *ΔH* for both series.

The scaling exponent (*ζ*(*q*)) obtained from MF-DFA for the four study areas showed a stronger multifractal character than the GSF results, especially for the NDVIa. For the NDVI, A1 had the most multifractal profile, with A2 and A3 on top and A4 as the least multifractal. This order shifted with the NDVIa, where all areas showed a more multifractal character, while A4 changed from the least to the most multifractal, right below A1 (Figure 9).

### 3.5. Sources of Multifractality

The shuffle and surrogate NDVI series analyses with MF-DFA revealed similar patterns across the study areas (Figure 10). Generally, all shuffle series had generalized Hurst exponents very close to 0.5, similar to a Brownian motion. The surrogate series reported generalized Hurst exponents with a smaller multifractality (*ΔH*) than the original series (Table 4). However, some differences in their patterns are worth mentioning. A1 and A2 presented a surrogate series that diverged more heavily from the original series. At lower-order statistical moments (*q* = 1, 2), the shuffle series was slightly lower than a typical Brownian motion. On the other hand, A3 and A4 surrogate series were very similar to the original series, and their shuffle series were almost always at 0.5.

The NDVIa results were similar to the mentioned NDVI results. However, the original and surrogate series presented a more antipersistent character, and the difference between the surrogate and the original series was more conspicuous for A3 and A4. Whilst similar to the NDVI, the A3 NDVIa presented a more divergent pattern between these series in higher-order statistical moments (q > 2). A4 presented a more significant difference in the smaller-order statistical moments (q < 2). To analytically examine the influence of both sources of multifractality, Hcor and Hpdf were calculated (Table 5). Study areas present a higher dominance of long-range correlation multifractality, although A1 presented a mix of both sources with Hpdf > 0.1.

## 4. Discussion

NDVIa series were used and compared to NDVI series in order to study the effect of seasonality removal on the trends present in the NDVI. The temporal trends were still visible using the NDVIa. Their presence in the anomaly series implies that there is an inherent trend to the NDVI series. This trend may be due to changes in land use or climate change effects worthy of further consideration. As a result of the similarities between the NDVI and NDVIa, we compared fractal analysis and multifractal analysis. The HI and *H*(*2*) from the GSF provided evidence of a high persistence in the study areas for both series, NDVI and NDVIa. With A1 showing a minor persistent character in all cases. This difference may be related to the characteristics and less-diverse vegetation of this area, where no tree cover or woodland exists.

Nonetheless, some differential patterns were observed between the NDVI and NDVIa for HI. When we removed the seasonality and analyzed the NDVIa series, HI revealed that persistence increased for all study areas compared to the NDVI. In contrast, the NDVIa showed less persistence than the NDVI when using *H*(*2*) extracted from the GSF. This opposite behavior is probably due to the normalization that the NDVI and NDVIa series suffer in the HI calculations, removing seasonality for both indices. This was not seen for the GSF and MF-DFA.

The use of generalized Hurst exponents based on MF-DFA resulted in lower values in all study areas. When NDVI series were used, only Area A4 showed persistence, while A2 and A3 revealed no or little long-term correlation and A1 exhibited an antipersistent character. The decrease in generalized Hurst exponents values was exacerbated when the NDVIa was used, with values suggesting a robust antipersistent character for all study areas.

Different time scales were used for the GSF and MF-DFA. Four lag scales were used for the GSF from 8 to 64 days. On the other hand, up to five time scales were used for MF-DFA, from 32 to 512 days. Given that these two analyses shared only the time scales of 32 and 64 days, their results must be compared cautiously. The R/S Hurst exponent and GSF *H*(*2*) exhibited persistent character for both indices and all study areas. However, when we applied the NDVIa instead of NDVI, the *H*(*2*) values decreased for the GSF. While the GSF generalized Hurst exponents for the NDVI series showed three different patterns, one for A1, another for A2 and A3, and a last one for A4, this clustering shifted when using the NDVIa. All study areas showed less-persistent characters, and two patterns appeared: A1 and A2 had a stronger dependency on q while A3 and A4 were almost constant as q grew. The drop in persistence (*H*(*2*)) between NDVI and NDVIa was also visible with MF-DFA.

The MF-DFA NDVI generalized Hurst exponents all started above 0.5 (i.e., as antipersistent), but A1–A3 dropped below 0.5 after *q* = 2. A4 had a similar pattern but with constant values. With the NDVIa, all of them started almost at or below 0.5. Concerning the NDVIa series, the MF-DFA generalized Hurst exponents were more similar to NDVI exponent values than in the case of the GSF. Simultaneously, the MF-DFA gave a more antipersistent character, and the GSF showed a stronger multifractality with larger *ΔH* values. MF-DFA results were in agreement with similar study areas with degraded vegetation [36]. The MF-DFA *H*(*q*) seemed to single out A4 on the top (the area with minor human influence), while A2 and A3 (a mix of grassland, cropland, shrubland, and some forested areas) were intertwined together. Below all of them appeared A1, being an area covered mainly by cereal crops. The somewhat similar land use in A2 and A3 resulted in more similar behavior than A1. A difference between herbaceous and tree-dominated areas was also seen in the slope of the NDVIa for *H*(*q*). A1 and A2 (herbaceous = −dominated areas) presented a less persistent and steeper slope, while A3 and A4 (tree crop and forested areas) showed a more antipersistent profile but with a lower slope.

Differences in multifractality between the NDVI and NDVIa were observed with *ΔH* and the scaling exponent patterns. The results indicate that part of the multifractality is due to seasonality. Nevertheless, long-range correlations affected the multifractality observed with the NDVI and NDVIa surrogate series (i.e., before and after removing seasonality). On the other hand, there was a limited impact of broad probability density function for all study areas. The NDVIa presented a more decisive influence from the broad probability density function than the NDVI, particularly in A1, where its multifractality based on the broad probability density function of the NDVI series was higher than in other study areas. Therefore, we can appreciate two types of long-range correlations based on (i) seasonal and (ii) annual or longer memory processes. We found a greater difference in *ΔH* between the original series of the NDVI and NDVIa, compared to the difference between the original and respective surrogate for each index (NDVI and NDVIa). This larger difference between the NDVI and NDVIa indicates that seasonality had a decisive influence on overall multifractality in all areas, except for A1. In this case, the NDVIa series presented more multifractality than its surrogate NDVI series. This is likely due to the almost complete herbaceous nature of A1. Vegetation in A1 is entirely rainfed and cannot take full advantage of deeper soil water than arboreal vegetation. Therefore, its behavior is more dependent and similar to the precipitation, which is more dependent on the broad probability density function, unlike air temperature or humidity [39]. Area 2, while it is also dominated by herbaceous crops, presents a grassland area with few shrubs and some reforested pixels that will maintain greener vegetation when precipitation is scarce.

The generalized Hurst exponents primarily diverged between the GSF and MF-DFA. The GSF characterized the series as low-level multifractal and persistent. However, when MF-DFA was applied, a larger multifractality and an antipersistent character were observed for the same series. MF-DFA allows us to study the sources of multifractality and to distinguish among the rangelands. The NDVIa especially clearly clustered our areas into two groups: A1 and A2, the most herbaceous crop areas; and A3 and A4, with more considerable tree coverage. The different behavior of surrogate and original series can distinguish the mentioned groups, but it can also differentiate among each area. More areas should be studied to see if these analyses can be adequately used to characterize more types of rangelands. In this study, we used the average of the NDVI and NDVIa to study each area’s vegetation behavior as one unit. However, we aim to study each pixel’s behavior to research the spatial variability in the study areas. The use of robust statistical methods such as MF-DFA, NDVI, and NDVIa could characterize rangeland types and uses, which will enable a much more accurate characterization to inform land use and management optimization.

## 5. Conclusions

When comparing NDVI and NDVIa time series, we found similar trends in all study areas, which provides evidence of an inherent trend caused by land-use changes or climate change effects. When this inherent trend is present, MF-DFA allows us to study multifractality once the local trend is subtracted, and shows a multiscaling pattern, whilst the GSF can be affected by the inherent trend. Our analysis produced similar results to previous research conducted in semi-arid areas with analogous land uses and vegetation degradation [36]. Using the NDVI, the MF-DFA *H(q)* showed that the area with herbaceous crops had a slight antipersistent character. The areas with open forest presented a more persistent character, while those with mixed uses appeared in the middle. Examining the MF-DFA *H(q)* for the NDVIa, we could discriminate among different land types, as those that are herbaceous and more heavily cropped had a steeper slope, resulting in a higher *ΔH*. Simultaneously, those with tree coverage, whether it is an open forest or a tree crop, showed a more antipersistent *H(q)* but a smaller *ΔH* than then herbaceous-dominated areas. The use of surrogate and original series for the NDVIa produced different patterns for each study area. MF-DFA has been proven to enhance our skills in monitoring and discriminating among different land types for rangelands, supporting more accurate land use and management optimization. This paper focused on the global description of each study area, and further work should focus on the spatial heterogeneity of each area. Our approach provides relevant information on vegetation dynamics that can inform policymakers and assist the design of risk management programs such as index insurance systems.

## Figures and Tables

**Figure 1 entropy-23-00576-f001:**
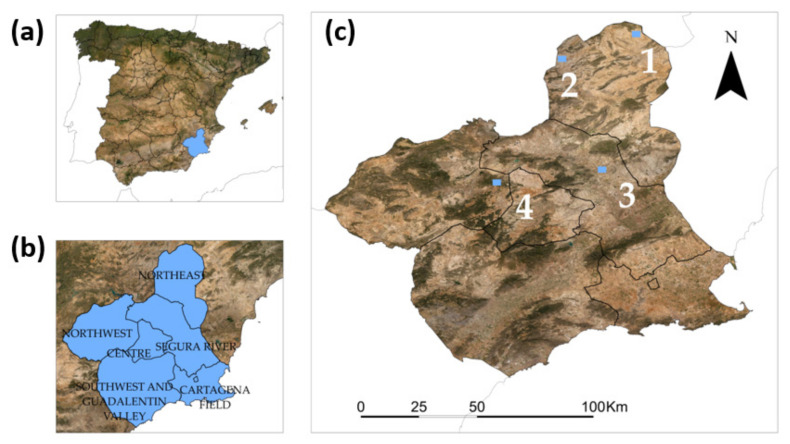
Location of the study area. (**a**) Autonomous Community of Murcia (Spain). (**b**) Agricultural regions of Murcia. (**c**) Study areas in three agricultural regions of Murcia. Numbers refer to the sampling areas. Source base map: Invierno 2020. Gobierno de España y Comunidad Autónoma de Murcia. CC-BY 4.0 scne.es 2020.

**Figure 2 entropy-23-00576-f002:**
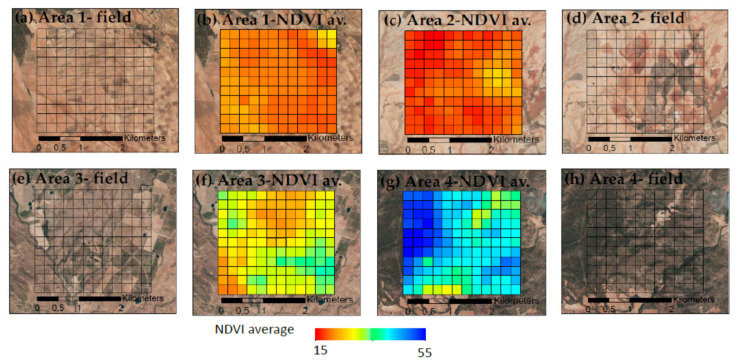
The four study areas, presenting the disposition of each pixel with a satellite image and its average NDVI values for all years: A1, rainfed croplands (**a**,**b**); A2, rainfed croplands+scrublands (**c**,**d**); A3, grassy+forested+crops (**e**,**f**); and A4, woodland+crops (**g**,**h**). Sources: Esri, DigitalGlobe, GeoEye, i-cubed, USDA FSA, USGS, AEX, Getmapping, Aerogrid, IGN, IGP, swisstopo, and the GIS User Community.

**Figure 3 entropy-23-00576-f003:**
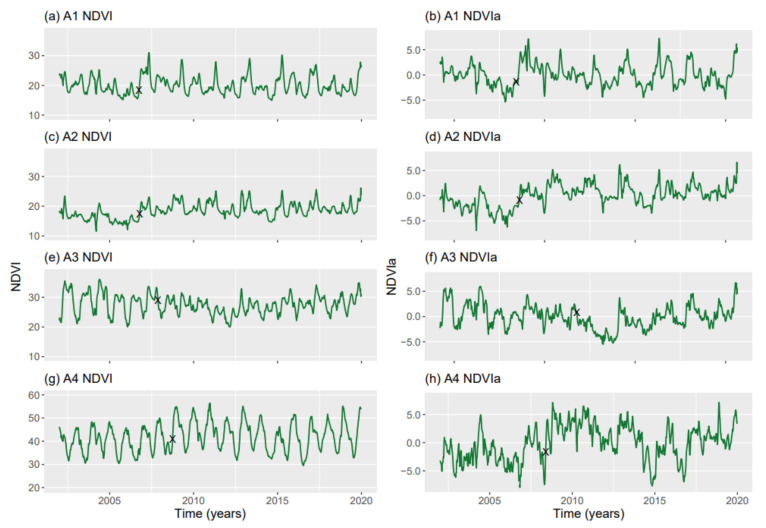
Series of average NDVI (**a**,**c**,**e**,**g**) and NDVIa (**b**,**d**,**f**,**h**) values for A1, rainfed croplands; A2, rainfed croplands+scrublands; A3, grassy+forested+crops; and A4, woodland+crops. The X symbol represents the point where the Pettitt test indicated a trend shift.

**Figure 4 entropy-23-00576-f004:**
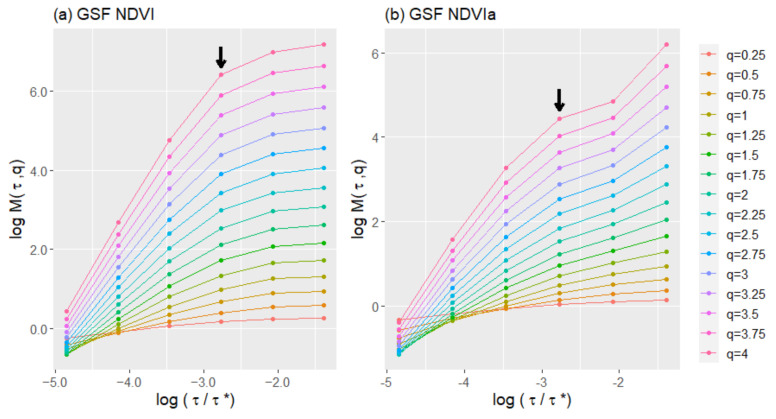
Log–log plot of moments function (*M(τ,q*)) versus lag time scale (*τ*) normalized with the maximum lag time scale (*τ**) for the generalized structure function (GSF), with q ranging from 0.25 to 4 for the NDVI (**a**) and NDVIa (**b**) of A3, grassy+forested+crops. The arrow marks the last point included in the linear regression. All the regressions obtained an R^2^ ≥ 0.97.

**Figure 5 entropy-23-00576-f005:**
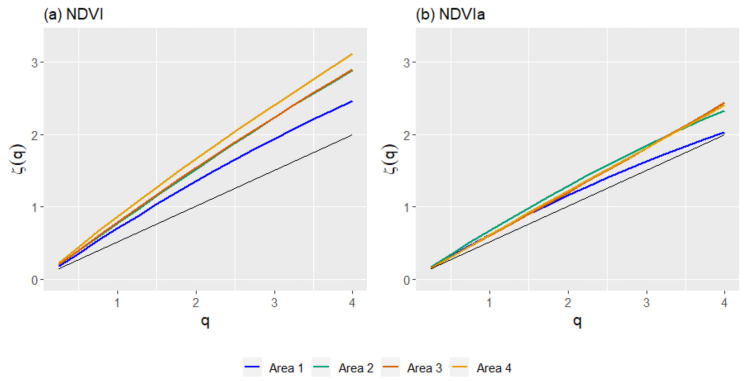
Generalized structure function (GSF) scaling exponents *ζ(q*) for the NDVI (**a**) and NDVIa (**b**) for the study areas: A1, rainfed croplands; A2, rainfed croplands+scrublands; A3, grassy+forested+crops; and A4 woodland+crops. In black, the scaling exponent of a Brownian motion.

**Figure 6 entropy-23-00576-f006:**
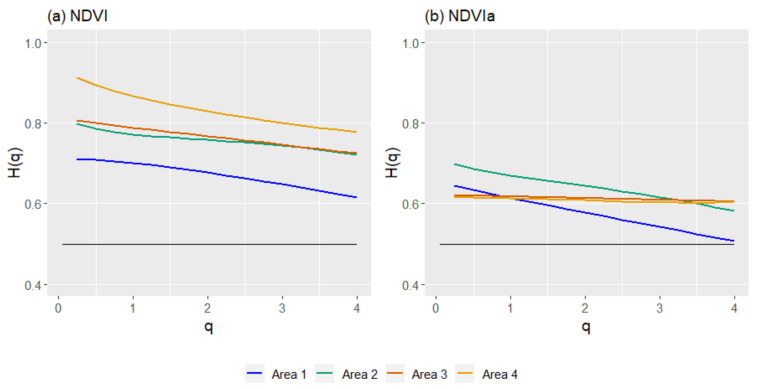
Generalized Hurst exponents *H*(*q*) for the study areas from the generalized structure function (GSF), for the NDVI (**a**) and NDVIa (**b**) for each area: A1, rainfed croplands; A2, rainfed croplands+scrublands; A3, grassy+forested+crops; and A4 woodland+crops. In black, the *H*(*q*) of a Brownian motion for comparison.

**Figure 7 entropy-23-00576-f007:**
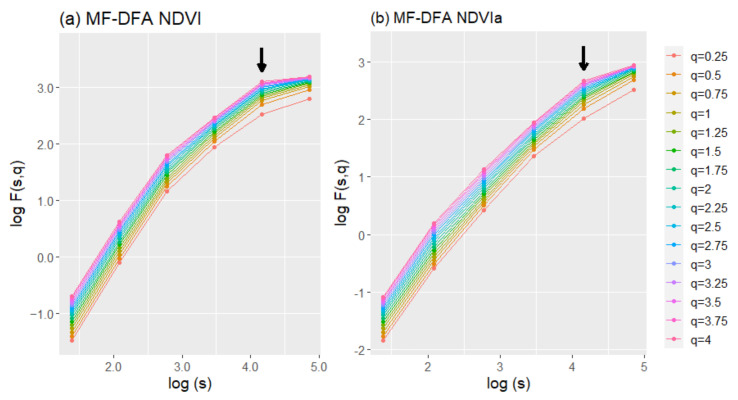
Log–log plots of fluctuation function (*F*(*s,q*)) versus time scale (*s*), for multifractal detrended fluctuation analysis (MF-DFA), with q ranging from 0.25 to 4 for the NDVI (**a**) and NDVIa (**b**) for A2 (rainfed croplands+scrublands). The arrows mark the last points included in the regression lines. All the regressions obtained an R^2^ ≥ 0.97.

**Figure 8 entropy-23-00576-f008:**
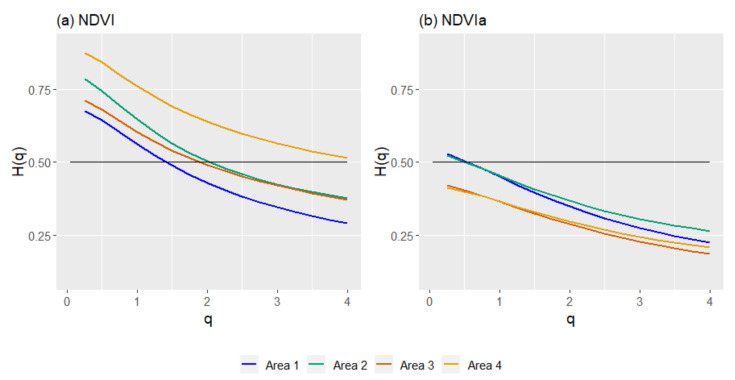
Generalized Hurst exponents (*H*(*q*)), from the multifractal detrended fluctuation analysis (MF-DFA), for the NDVI (**a**) and NDVIa (**b**) for each area: A1, rainfed croplands; A2, rainfed croplands+scrublands; A3, grassy+forested+crops; and A4 woodland+crops. In black, the *H*(*q*) of a Brownian motion for comparison.

**Figure 9 entropy-23-00576-f009:**
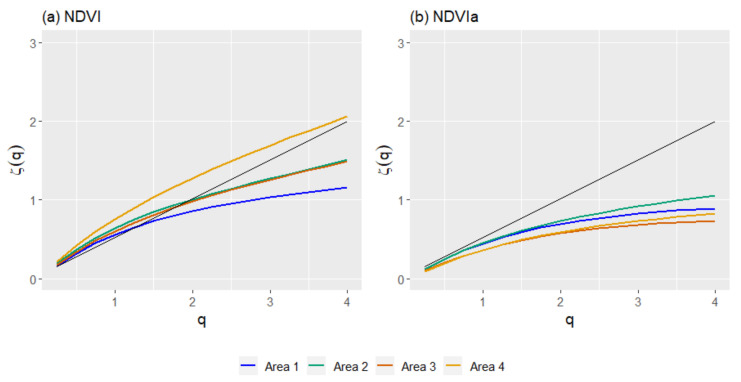
Scaling exponents (*ζ*(*q*)) from multifractal detrended fluctuation analysis (MF-DFA), for NDVI (**a**) and NDVIa (**b**), for the study areas: A1, rainfed croplands; A2, rainfed croplands+scrublands; A3, grassy+forested+crops; and A4 woodland+crops. In black, the scaling exponent of a Brownian motion for comparison.

**Figure 10 entropy-23-00576-f010:**
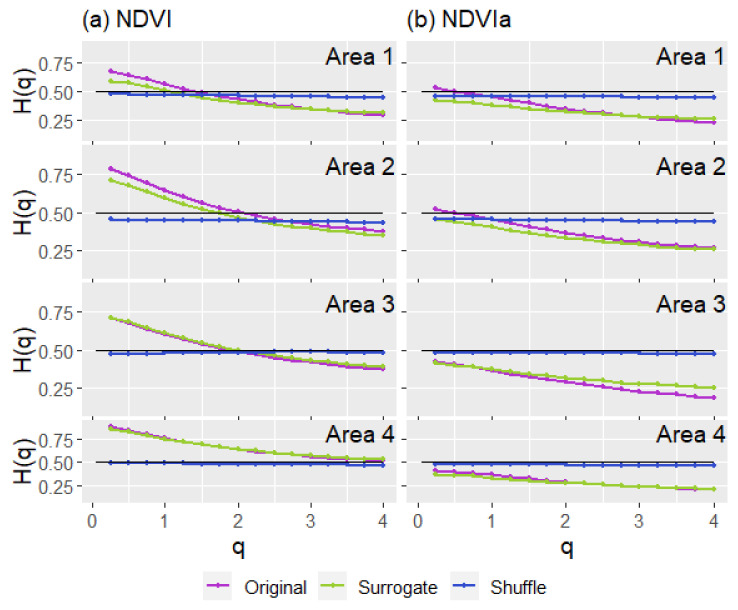
Generalized Hurst exponents (*H*(*q*)) from multifractal detrended fluctuation analysis (MF-DFA), for NDVI (**a**) and NDVIa (**b**), for the study areas: A1, rainfed croplands; A2, rainfed croplands+scrublands; A3, grassy+forested+crops; and A4 woodland+crops. The original series is in dotted purple, the surrogate series is in dotted green, and the shuffle series is in dotted blue. To compare the figures, the *H*(*q*) of a Brownian motion is in solid black.

**Table 1 entropy-23-00576-t001:** Mann–Kendall test results for the four study areas: A1, rainfed croplands; A2, rainfed croplands+scrublands; A3, grassy+forested+crops; and A4 woodland+crops. The same results were obtained using NDVI and NDVIa. All results were statistically significant.

Area	Trend	Kendall’s tau	*p*-Value
A1	Decreasing	−0.04	<0.05
A2	Increasing	0.26	<0.05
A3	Decreasing	−0.05	<0.05
A4	Increasing	0.08	<0.05

**Table 2 entropy-23-00576-t002:** Corrected R/S Hurst indices (HIs), generalized Hurst exponents for *q* = 2 (H(*2*)) based on generalized structure function (GSF) and multifractal detrended fluctuation analysis (MF-DFA) for the original NDVI and NDVIa from different study areas: A1, rainfed croplands; A2, rainfed croplands+scrublands; A3, grassy+forested+crops; and A4 woodland+crops.

Area	HI		GSF		MF-DFA	
	NDVI	NDVIa	NDVI	NDVIa	NDVI	NDVIa
A1	0.684	0.736	0.677	0.578	0.430	0.348
A2	0.863	0.893	0.758	0.644	0.504	0.367
A3	0.762	0.845	0.767	0.614	0.490	0.287
A4	0.728	0.907	0.829	0.608	0.638	0.295

**Table 3 entropy-23-00576-t003:** Multifractality strength based on *ΔH*, using the generalized structure function (GSF) and multifractal detrended fluctuation analysis (MF-DFA), for all original NDVI and NDVIa series from different study areas: A1, rainfed croplands; A2, rainfed croplands+scrublands; A3, grassy+forested+crops; and A4, woodland+crops.

Area	GSF		MF-DFA	
	NDVI	NDVIa	NDVI	NDVIa
A1	0.094	0.137	0.386	0.304
A2	0.063	0.116	0.409	0.259
A3	0.075	0.015	0.338	0.236
A4	0.116	0.012	0.360	0.206

**Table 4 entropy-23-00576-t004:** Multifractal strength measure by *ΔH* of multifractal detrended fluctuation analysis (MF-DFA) for original series (NDVI and NDVIa), surrogate series (NDVI_su and NDVIa_su), and shuffle series (NDVI_sh and NDVIa_sh). Study areas: A1, rainfed croplands; A2, rainfed croplands+scrublands; A3, grassy+forested+crops; and A4, woodland+crops.

Area	NDVI	NDVI_su	NDVI_sh	NDVIa	NDVIa_su	NDVIa_sh
A1	0.386	0.284	0.029	0.304	0.172	0.017
A2	0.409	0.361	0.019	0.259	0.198	0.017
A3	0.338	0.317	0.001	0.236	0.162	0.014
A4	0.360	0.326	0.017	0.206	0.157	0.013

**Table 5 entropy-23-00576-t005:** Difference between the *ΔH* of the original and surrogate series (Hcor) and the difference in the *ΔH* of the original and shuffle series (Hpdf) for the NDVI and NDVIa for the study areas: A1, rainfed croplands; A2, rainfed croplands+scrublands; A3, grassy+forested+crops; and A4, woodland+crops.

Area	Hcor		Hpdf	
	NDVI	NDVIa	NDVI	NDVIa
A1	0.357	0.287	0.103	0.132
A2	0.390	0.241	0.048	0.061
A3	0.337	0.222	0.021	0.074
A4	0.343	0.193	0.035	0.049

## Data Availability

The data presented in this study are available on request from the corresponding author.

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
