# Peer review of "Generalized Structure Functions and Multifractal Detrended Fluctuation Analysis Applied to Vegetation Index Time Series: An Arid Rangeland Study"

_entropy, 2021, doi:10.3390/e23050576_

Round 1

Reviewer 1 Report

This paper explored the scaling character of NDVI and NDVI anomaly (NDVIa) time series by applying Hurst index (HI), Generalized Structure-function (GSF), multifractal detrended fluctuation analyses (MF-DFA). Overall, the paper was well written with clear research objectives, reasonable organizational framework, sufficient analysis and discussion. The biggest problem is that the method is not clear enough, some terms are confused and difficult to distinguish. Frankly speaking, I'm not very familiar with these methods, but I try my best to understand them. Unfortunately, I don't feel that the description of the methods is enough to let the readers know exactly what the author did. The paper deserves a major revision before it can be published in Entropy.

Some specific comments:

  1. Lines 28-29 are difficult to be understood.
  2. The data source used in Figure. 2 should be added. It is difficult to understand the meaningful results and conclusions of the conclusions obtained by calculating the average value of the region rather than analyzing the time series based on the pixel or a certain type. There is a strong heterogeneity in land cover types from Figure 2. The pixels are mixed in MODIS data. So how will the mixed pixels affect your results? Also it is better to give the MODIS image.
  3. The paper mentioned generalised Hurst exponents, Hurst exponents, corrected Hurst index, they are difficult to distinguish if not given clear definition. Line 148-150 said use “nonlinearTseries” (version 0.1) package. You also mentioned that you used package "prama". I am a little confused what did you use.
  4. what do the authors want to say using "GSF are generalised correlation functions."? The method sections 2.4-2.6 are not well structured and described. I was really confused about these unclearly defined terms.
  5. Line 176 the purpose of ΔH used here should be described.
  6. Line 193 "study a line was chosen" is hard to understand.
  7. Line 203 used H(p), what's its difference with that of line 176. They should be clearly stated.
  8. Line 211-215 is difficult to understand. you have mentioned H(0.25)-H(4) in GSF method, did you mean it was also calculated in MF-DFA method?
  9. Line 215 Why did you calculate ζ(q) here? what's the difference with that of Line 169-170.
  10. Is the shift detected by NDVI and NDVIa the same? What's the use of the detected shift?
  11. There is no need to separate Figure 3 and Figure 4.
  12. Line 268 The scaling exponent was not introduced in the method part. As the authors' result and explanation, ζ(q) is the scaling exponent, so it should be clearly defined in the Line 168-171.
  13. Brownian motion and its function were not described in the method.
  14. General Hurst exponent can be calculated by GSF and MF-DFA, so it is necessary to make a good distinction in the paper. At present, it often takes a lot of time to identify and easily confuse the readers.
  15. Line 313 is not easy to understand. What is five (NDVIa)?
  16. Line 316-319 is puzzling. Although I can understand what the authors want to say from Figure 9.
  17. The expression of Lines 400-401 may not be very appropriate. The calculation method of NDVIa only reduces the influence of seasonal or phenological changes, and cannot remove the influence of trend. Line 449-450 has also been mentioned the seasonality.
  18. Line 405 Why is H(2) mentioned? There is no discussion in the paragraph, and there is a discussion in line412 with a “in contrast”?
  19. I think the feature of this paper is to use different methods to analyze time series NDVI and NDVIa, and the conclusion will be better to focus on this content. At present, it only focuses on MF-DFA and is more like a discussion.

Reviewer 2 Report

The paper entitled "Generalised Structure Functions and Multifractal Detrended Fluctuation Analysis applied to Vegetation Index Time Series: an arid rangeland study" it is suitable for publishing in Entropy Journal. The paper is original it is well explained ans structured. On the other hand the fractal and multfractal algorithm are applied in a correct way. Only some details are necessary to take in account: 

1)Figure 2: Add scale bar to each figure.  
2)Figure 3 and 4: write the unit time in the graphs  (years).

3) Review all variables in the text, all of them sould be in italics.

4) Conclusion section: Authors state in line 486-488 "our analysis produced similar results to other studies..." Please add the references of the authors you refer to in this section. 

5) Review the formatting of the references list. There is no space between de the publication year in most of them.

Round 2

Reviewer 1 Report

The authors responsed my concerns well, and I think this paper can be published in Entropy.